# Nighttime Mesospheric Ozone During the 2002 Southern Hemispheric Major Stratospheric Warming

Christine Smith-Johnsen<sup>1</sup>, Yvan Orsolini<sup>2,3</sup>, Frode Stordal<sup>1</sup>, Varavut Limpasuvan<sup>4</sup>, and Kristell Pérot<sup>5</sup>

<sup>1</sup>Department of Geosciences, University of Oslo, Norway

<sup>2</sup>Birkeland Centre for Space Science, University of Bergen, Norway

<sup>3</sup>Norwegian Institute for Air Research, Kjeller, Norway

<sup>4</sup>Coastal Carolina University, Conway, SC, USA

<sup>5</sup>Department of Earth and Space Sciences, Chalmers University of Technology, Gothenburg, Sweden

Correspondence to: Christine Smith-Johnsen (christine.smith-johnsen@geo.uio.no)

Abstract. A Sudden Stratospheric Warming (SSW) affects the chemistry and dynamics of the middle atmosphere. The major warmings occur roughly every second year in the Northern Hemispheric (NH) winter, but has only been observed once in the Southern Hemisphere (SH), during the Antarctic winter of 2002. Using the National Center for Atmospheric Research's (NCAR) Whole Atmosphere

- Community Climate Model with specified dynamics (WACCM-SD), this study investigates the effects of this rare warming event on the ozone layer located around the SH mesopause. This secondary ozone layer changes with respect to hydrogen, oxygen, temperature, and the altered SH polar circulation during the major SSW. The 2002 SH winter was characterized by three zonal-mean zonal wind reductions in the upper stratosphere before a fourth wind reversal reaches the lower stratosphere,
- marking the onset of the major SSW. At the time of these four wind reversals, a corresponding episodic increase can be seen in the modeled nighttime ozone concentration in the secondary ozone layer. Observations by the Global Ozone Monitoring by Occultation of Stars (GOMOS, an instrument on board the satellite Envisat) demonstrate similar ozone enhancement as in the model. This ozone increase is attributable largely to enhanced upwelling and the associated cooling of the alti-
- tude region in conjunction with the wind reversal. Unlike its NH counterpart, the secondary ozone layer during the SH major SSW appeared to be impacted more by the effects of atomic oxygen than hydrogen.

## 1 Introduction

In the winter hemisphere, a large temperature difference can be found between the dark polar region and the sunlit extratropics. Combined with the Coriolis effect, this meridional temperature gradient leads to the formation of the eastward Polar Night Jet (PNJ) at high latitudes in the stratosphere. The PNJ acts as a barrier against meridional transport, so the polar region confined by the jet is isolated

from the midlatitudes Holton (2004). Planetary waves originating in the troposphere can propagate up to the stratosphere, where they can break and reduce the PNJ strength. This wave breaking can even cause the PNJ to reverse and become westward, which is called Sudden Stratospheric Warm-

ings (SSWs) because of the associated warming in the stratosphere.

According to the World Meteorological Organization (WMO) definition McInturf (1978), an SSW event is considered major when the PNJ reverses at 60 degrees latitude at the 10 hPa pressure level,
along with a reversal of the meridional temperature gradient between 60 and 90 degrees latitude. The altered wind direction occurs frequently in the upper stratosphere and mesosphere, but does not always propagate down to 10 hPa. Major SSWs are common in the Northern Hemisphere, where they occur about every second year Charlton and Polvani (2007), though with large inter-annual and decadal variability. In the Southern Hemisphere, only one major SSW (in 2002) has ever been observed since SSWs were first discovered in the 1950s Scherag (1952).

In addition to affecting the large scale circulation, SSW events alter the vertical propagation of gravity waves. The westward PNJ during a major SSW prevents gravity waves with westward phase speeds from propagating upward from the troposphere into the mesosphere. During normal winter

- conditions, gravity waves dissipate in the Mesosphere and Lower Thermosphere (MLT) region, exerting a westward drag on the background wind. With the PNJ reversal throughout the stratosphere, only gravity waves with eastward phase speeds (exceeding the tropospheric eastward wind speed) are allowed to propagate upward into the mesosphere. Upon damping, these eastward gravity waves contribute to an anomalous ascent and cooling in the mesosphere Siskind et al. (2005), Limpasuvan
- and Richter (2012). In addition to the warming of the polar stratosphere and the cooling of the polar mesosphere, there are also model and observational evidences of a lower thermospheric warming following the reversal of the PNJ Funke et al. (2010), Liu and Roble (2002).
- Around the mesopause (at 90-95 km altitude), a secondary peak in the ozone density is found Hays
  and Roble (1973), Kaufmann et al. (2003), due to the cold temperature and the high atomic oxygen density at this altitude region Smith and Marsh (2005). High above the main stratospheric ozone layer (at 30 km altitude), nighttime ozone in this secondary layer is in photochemical equilibrium Smith and Marsh (2005). The main ozone source at this altitude is atomic oxygen, while the sinks are atomic hydrogen and atomic oxygen. The ozone production is negatively correlated with temperature, and the ozone density (given as molecules/cm<sup>3</sup>) can be expressed as:

$$[O_3] = \frac{c_1 \cdot [O] \cdot [O_2] \cdot M}{c_2 \cdot [H] + c_3 \cdot [O]} \tag{1}$$

The reactions rates (c1 - 3) and the air density M are all temperature dependent, with  $c_1 = 6 \cdot 10^{-34} (300/T)^{2.4}$ ,  $c_2 = 1.4 \cdot 10^{-10} exp(-470/T)$ ,  $c_3 = 8 \cdot 10^{-12} exp(-2060/T) cm^3 molecules^{-1}s^{-1}$ , and M = p/kT (where p is pressure and k is Boltzmann's constant) Sander et al. (2006). Because of

60 the short lifetime (on the order of minutes), nighttime ozone cannot be transported. Rather, it rapidly responds to transport of its longer-lived sources and sinks. Daytime ozone is quickly destroyed by sunlight, and is an order of magnitude smaller than in nighttime.

Tweedy et al. (2013) showed that a brief enhancement in nighttime ozone occurs in the secondary layer at high latitudes during the Northern Hemispheric major SSWs. This enhancement is mostly driven by cooling and the decrease in atomic hydrogen during the mesospheric ascent. To date, no studies have documented the variation in the secondary ozone during the lone major SSW in the Southern Hemisphere. To this end, the aim of this study is to document and explain, for the first time, a similar ozone enhancement during the Southern Hemispheric SSW that took place in 2002.

## 70 2 Methods, model and instrumentation

This study uses the National Center for Atmospheric Research (NCAR) Whole Atmosphere Community Climate Model (WACCM 4) to examine the secondary ozone layer during the major SSW. WACCM is a chemistry-climate model that reaches up to 150 km altitude, and is part of the Community Earth System Model (CESM) Marsh et al. (2013). In this study, WACCM is used with specified

- dynamics (hereafter referred to as WACCM-SD), such that horizontal winds, temperature and surface pressure are constrained to analyses from NASA Global Modeling and Assimilation Office Modern-Era Retrospective Analysis for Research and Applications (MERRA) Rienecker et al. (2011), by the method described in Kunz et al. (2011). This nudging is applied from the surface to about 50 km altitude (0.79 hPa), and the model is free running above 60 km (0.19 hPa), with a linear transition in
- between. Simulated from 1990 until 2010, the model has a horizontal resolution of 1.9° latitude and 2.5° longitude, and a variable vertical resolution with 88 vertical levels. The model output is only once a day (00:00 GMT). We examine in detail the austral winter and spring 2002, and compare it to the climatology over the years 1990-2010 (excluding 2002).
- Although we are interested in ozone *variability* in this paper, hence model biases are not so critical, it is worth noting that WACCM tends to underestimate the mesospheric ozone abundance. In comparing ozone from WACCM4 with measurements by the Sounding of the Atmosphere using Broadband Emission Radiometry (SABER) on the Thermosphere-Ionosphere-Mesosphere Energetics and Dynamics (TIMED) satellite, Smith et al. (2014) showed that ozone volume mixing ratios
- were 50 percent lower than observed. They pointed out that this difference is likely due to negative bias in atomic oxygen. Too little atomic oxygen is transported down from the MLT region into the

mesosphere as a result of weak vertical diffusion and eddy mixing.

We furthermore examine the secondary ozone layer during the Antarctic winter and spring 2002
using observations of the Global Ozone Monitoring by Occultation of Stars (GOMOS). GOMOS is a stellar occultation instrument on board the European platform ENVISAT, launched in March 2002 and in operation until April 2012 (see Bertaux et al. (2010) for an overview of the mission). Ozone profiles are retrieved from measurements in the UV-visible spectral range, using the stellar occultation method, from the tropopause to the lower thermosphere. The vertical resolution in the mesosphere is 3 km. The retrieval technique is described by Kyrölä et al. (2010) and an analysis of

retrieval errors is presented by Tamminen et al. (2010). We use only nighttime measurements in our study. In order to ensure a good quality of the mesospheric ozone profiles, it is important to filter the data based on the star effective temperature. Following European Space Agency's recommendation ESA (2007), we screen out all profiles from cold stars (temperature lower than 6000 K).

Finally, our results will be compared to the similar study by Tweedy et al. (2013) of the Northern Hemisphere SSWs. The goal is to determine if the Southern Hemispheric SSW affects the secondary ozone layer in a different way than in the Northern Hemisphere.

## 3 Results and Discussion

## 110 3.1 The sudden stratospheric warming's effect on the secondary ozone layer

The zonal-mean zonal winds at  $60^{\circ}$ S in 2002 can be compared to the climatology over the period 1990-2010 (Fig. 1, top row). The climatological eastward PNJ stretches from  $10^{2}$  hPa to  $10^{-2}$  hPa in August, while westward winds prevail above. The eastward winds progressively weaken as spring transitions to summer in September and October, while westward winds descend to the lower meso-

- sphere. In 2002, the zonal-mean zonal wind is reduced in strength three times in the mesosphere (down to 1 hPa), before a fourth stronger wind reversal propagated further down, reaching 10 hPa. This marks the onset of the major SSW (based on the WMO definition), on 25th of September 2002. In the upper left panel of Fig. 1, all four wind reversals are highlighted with vertical lines. After the onset, the PNJ recovery to eastward direction is only found above  $10^{-1}$  hPa, while between this level
- and  $10^1$  hPa, westward winds persisted into the summer. We also note that the mesospheric zonal winds returned to the eastward direction following the three previous reductions.

Associated with the PNJ reversal, we expect a change in the vertical transport. The second row of panels in Fig. 1 shows the the residual vertical velocity, averaged between 55°S and 75°S, again

for 2002 (left) and for the climatology (right). At altitudes around pressure 1 hPa to  $10^{-2}$  hPa, a strong climatological descent is dominant until mid September, followed by a weak ascent at the

summertime transition around early October. In 2002, regions of ascent can be seen at several pressure levels between  $10^{-1}$  hPa to  $10^{-3}$  hPa for each of the three first zonal wind reduction episodes. During the fourth reversal, the ascent is strongest and covers a large vertical range (pressure between  $10^{0}$  hPa and above at  $10^{-2}$  hPa). The strong descent in the upper mesosphere above is also enhanced

$10^0$  hPa and above at  $10^{-2}$  hPa). The strong descent in the upper mesosphere above is also enhanced after the SSW onset (solid vertical line), as seen by Kvissel et al. (2012).

These strong vertical motions will in turn affect the zonal-mean temperature, as shown in the third row of Fig. 1 for the same latitude band average. Repeated cooling episodes (associated with adiabatic ascent) can be seen below the cold mesopause region at the time of the wind reversals, showing

a downward propagation with time. The SSW onset gives rise to the strongest and most persistent cooling, starting at  $10^{-3}$  hPa and propagating down to  $10^{-1}$  hPa. The strong upper mesospheric descent that begins at the same time above  $10^{-2}$  hPa causes a warm anomaly in this region and limits the cooling.

The bottom row in Fig. 1 shows zonal-mean ozone volume mixing ratio for the same latitude band. The secondary ozone layer is situated at the mesospause around  $10^{-3}$  hPa (or an altitude of about 90-95 km), as seen in 2002 (left) and the climatology (right). Since the ozone in the mesopause region has a much higher volume mixing ratio at night compared to daytime, the climatological ozone

- decreases towards spring as the nights get shorter and insolation increases. At the time of the wind reversals and mesospheric coolings in 2002, increases can be seen in the secondary ozone layer. By early October 2002, the upper mesosphere warming associated with the strong descent contributes to lower ozone, since the latter is anti-correlated with temperature. At these austral high latitudes, the volume mixing ratio of the zonal-mean ozone in the secondary layer is around 2 ppmv, while the
- nighttime values are of the same order of magnitude as in the stratospheric ozone layer. This will be illustrated below as we consider the nighttime sector only.

The zonal wind reversals mark strong ozone departures from the climatology in the mesopause region. Fig. 2 show ozone, temperature, atomic oxygen (the main ozone source, but also sink) and atomic hydrogen (the main ozone sink) at the mesospause in 2002 (solid line). The corresponding climatology (dashed line, with the gray lines indicating one standard deviation) is also illustrated. The plots reveal how the mesospheric composition during the repeated strong coolings (and ascents) in 2002 differs from other years. The ozone volume mixing ratio is correspondingly higher than the climatological condition in August and September of 2002, with short-duration peaks at the time of

160 the wind reversals (shown again as vertical lines). On the contrary, temperature shows pronounced decreases at the time of the wind reversals. Atomic oxygen shows weak decreases, within the standard deviation of the climatology. Hydrogen exhibits weak decreases superposed on a background seasonal decline, and its abundance is lower than the climatology through the entire period. The

largest decrease in temperature, atomic oxygen and hydrogen occurs the 26th of September, one day after the SSW onset. Following the decrease, there is a strong increase in temperature, oxygen, and hydrogen at  $10^{-3}$  hPa, associated with the strong upper mesospheric descent.

In order to better represent spatial variations, polar maps at the mesopause altitude (pressure  $10^{-3}$  hPa) of ozone, temperature, atomic oxygen and hydrogen are shown for a few days before the SSW

- onset Fig. 3, and for the onset day (September 25th) in Fig. 4. Before the SSW, high values of atomic oxygen and hydrogen can be seen in the regions with high temperature. When the temperature decreases, at the SSW onset, so does the atomic oxygen and hydrogen. In the nighttime sector at latitudes 55°-75°S, the enhancement in ozone can clearly be seen in conjunction with this low temperature, and the low values of atomic hydrogen (the main ozone sink) and atomic oxygen (the main
- ozone source, but also sink). Only nighttime ozone is shown, with masked daytime region values. The ozone volume mixing ratio reaches 8 ppmv in the nighttime, but is close to zero in the daytime, explaining why the the zonal mean is around 3 ppmv (see Figure 2). In the next section, the focus will be on just the nighttime sector. Although our focus is on the high latitudes, the polar maps also show ozone enhancements at mid latitudes, associated with cold temperatures and low abundances of atomic hydrogen and oxygen.
- of atomic hydrogen and oxygen.

## 3.2 Mechanisms behind the nighttime ozone increase

As discussed in the introduction, nighttime ozone in the MLT region is in chemical equilibrium. Its concentration is only dependent on temperature, on the concentrations of atomic and molecular oxygen, and of atomic hydrogen, as given by Equation 1. We can use the local temperature and densities of hydrogen and oxygen from WACCM-SD to calculate the nighttime ozone, and determine if the model ozone is in chemical equilibrium. Upper panel in Fig. 5 shows nighttime ozone from WACCM-SD (black line), and nighttime ozone calculated according to Equation 1, (gray line), after averaging zonally and over the 55°S-75°S latitude band. We see an excellent agreement between this calculated chemical equilibrium nighttime ozone, and the model nighttime ozone.

This relation can be used to illuminate which factors in the photochemical equilibrium (temperature, hydrogen, or atomic oxygen) are most important. We let one of them vary at the time, while the other factors are fixed to their mean values over the whole period, following Tweedy et al. (2013). To see the effect of temperature upon nighttime ozone, Equation 1 is used with a time-mean value for

hydrogen and atomic oxygen for the period, while only the nighttime temperature is varied. The resulting nighttime ozone estimate is shown as the blue line in Fig. 5. The same procedure is followed to see the effect from varying hydrogen (green line) and from atomic oxygen (red line). The result is that the brief changes in temperature (coolings) during the episodic wind reversals account for most of the increase in ozone. The oxygen increases have however a counteracting effect: the tem-

perature alone would cause a stronger ozone increase than actually seen in the model. The hydrogen decreases during the reversals only contribute to a weak ozone increase, but the slow seasonal hydrogen decline would alone cause a slow build-up of the ozone. During the upper-mesospheric descent that follows the SSW onset, ozone is declining due to the warm temperatures, but this effect is first counteracted by the brief pulse of atomic oxygen.

## 205 3.3 GOMOS satellite observations

GOMOS regular observations began in mid-August 2002. Our study is based on the very first measurements of the mission, which explains the gaps in the data set (September 8 to 16, and October). Approximately 570 ozone profiles (after screening to retain only occultations of bright stars) have been retrieved from mid-August to late September 2002 in the 55-75°S latitude band. 99% of the

- profiles correspond to a solar zenith angle higher than 100 °, hence to nighttime conditions at meso-spheric altitudes. Figure 6 shows the evolution of the daily zonal mean ozone number density during this time period, at the altitude corresponding to the peak of the secondary ozone layer (88 km on average). The shaded area represents the statistical error (equal to the 95% confidence interval of the daily zonal mean). The estimated retrieval errors are not shown here, but they are much smaller
- than the natural variability. The average of the daily zonal mean ozone density measured in 2003 and 2004, during the same time of the year, is shown in gray for comparison. The instrument was affected by a technical problem in 2005, which resulted in a change in the occultation scheme after this year. We decided to use only the measurements made before this malfunction in our study, to ensure a temporal and geographical sampling similar to 2002.

Several peaks in ozone number density clearly appear in Figure 6. 35% and 40% increases, compared to the 2003-2004 average, are observed around August 24 and September 1, respectively. The largest enhancement is observed in the end of September, when the measured ozone number density was approximately 60% higher in 2002 than in the other years. The ozone peak, as observed by GOMOS,

varies between 87 and 91 km during this period (not shown). GOMOS was then measuring mostly before sunrise, between 3AM and 6 AM local time. The varying local time sampling throughout the period may cause aliasing of tides, which modulate the height of the secondary ozone layer. After September 22nd, the number of GOMOS measurements is quite small, fluctuating between 2 and 15. This may also contribute to the lack of exact day-to-day correspondence with the WACCM-SD

zonal and meridional means of ozone (e.g. on Fig. 2, or on WACCM-SD ozone densities, not shown).

To make a direct comparison with the model results, we would need to consider temporal and geographical collocations with GOMOS measurements. However, as explained in the Methodology, WACCM-SD outputs used in this study are available only once a day (00:00 GMT). Furthermore, a

235 direct comparison of WACCM-SD and GOMOS with temporal and geographical collocations, while

in principle possible, would hence also fold tidal signals in the ozone evolution. However, Figure 6 shows that the phenomenon described in our model study has also been observed by a satellite remote sensing instrument: increased ozone densities in the secondary ozone layer have indeed been measured by GOMOS at southern high latitudes during the winter 2002 around the time of the wind reversals. Values were highest during the last week of September, when the middle atmosphere was

## 240 reversals. Values were hig affected by a major SSW.

#### 4 Conclusions

In their study of nighttime mesospheric ozone during Northern Hemisphere SSWs with WACCM-SD, Tweedy et al. (2013) concluded that the leading factors contributing to the ozone peak were

245 the decrease in temperature, followed by the decrease in hydrogen. In comparison, our study on the Southern Hemisphere case finds that atomic oxygen plays a larger role than hydrogen. This difference may be attributed to the influence of the seasonal cycle, given that the Southern Hemispheric SSW occurred later in the year than the mid-winter warmings studied in Tweedy et al. (2013). Also, the seasonal cycle in atomic oxygen is more pronounced in the Southern Hemisphere with a more 250 rapid decline in late winter/spring, lowering the production of ozone.

This paper focuses on the behavior of the secondary ozone layer during the Southern Hemispheric sudden stratospheric warming (SSW) in 2002. Using the Whole Atmosphere Community Climate Model with specified dynamics (WACCM-SD), we conclude that the polar mesospheric cooling

- above the stratospheric warming contributes mainly to the short-duration peaks in nighttime ozone. The concurrent decrease of atomic oxygen can reduce the effect of temperature change, while changes in hydrogen play a minor role. Our results agree with studies of the nighttime mesospheric ozone in the northern hemisphere. However, the effect from atomic oxygen is much stronger than what is found for the Northern Hemisphere, where Tweedy et al. (2013) concluded that hydrogen play a much stronger role.
- played a much stronger role.

Observations by the Global Ozone Monitoring by Occultation of Stars (GOMOS) instrument on the Envisat satellite confirms the ozone increase at the mesopause during the sudden stratospheric warming in 2002 although observational sampling makes direct comparison to the simulated ozone increases difficult.

 Acknowledgements. This research has been funded by the Norwegian Research Council through project 222390 (Solar-Terrestrial Coupling through High Energy Particle Precipitation in the Atmosphere: a Norwegian contri bution)

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
