# Peer review of "Nighttime Mesospheric Ozone During the 2002 Southern Hemispheric Major Stratospheric Warming"

_Atmospheric Chemistry and Physics, 2016_

## Short Comment (SC1) · 29 Sep 2016

This is an interesting paper and it is good to revisit this particular event which was, and still is, quite unusual. My main comment is that I think they could and should do a bit more to validate their model. One paper that is probably of relevance is that of Coy et al., (2005) "Modeling the 2002 minor warming event", [GRL, L 07808, doi: 10.1029/2005GL022400]. That presented what, to my knowledge, remains the only simulation of the mesospheric response to the SH 2002 warming. I think this should be cited. Admittedly, they covered the minor warming precursor in August, not the major event in September. This is because SABER yawed away from the SH and did not observe the major event. Nonetheless, the August period is covered by the present

authors' simulations and figures. For reference, I have attached Figure 3 from the Coy paper. Note that Coy et al show the vertical structure of the event up to the upper mesosphere. I suggest it would be useful for a similar analysis to be done here to help validate their modeled mesospheric response.

My other question concerns the hydrogen. They refer to a climatology in their figure. Is this climatology from WACCM itself? Because SABER did measure hydrogen during the August event. A paper by Mlynczak et al, JGR, 2014 documents the SABER H product. I believe this is an underutilized dataset and the present paper offers an ideal opportunity to compare their model with data.

Finally, and I'm sure they've noticed already, but somebody misspelled stratospheric in their title. (start) It seems correct on the PDF, but incorrect on the web site.

[Figure]

**Figure 3.** Zonal mean temperature changes (K) at 80°S as a function of time (18−28 August 2002) and pressure for a) NOGAPS-ALPHA 5-day forecast without RF, b) NOGAPS-ALPHA 5-day forecast with RF, and c) SABER observations. The contour interval is 2.5K. The NOGAPS-ALPHA forecasts have been smoothed with time using a 12 hr boxcar average and are not plotted at the beginning and end. Daily averaged SABER observations are plotted at 12 UTC.

**Fig. 1.**

---

## Referee Comment (RC1) · Anonymous Referee #1 · 31 Oct 2016

General comments

The manuscript addresses a topic of high scientific interest for knowing the state and variability of the mesosphere, in particular, the perturbations of the O3 secondary maximum distribution to a major SSW in the Southern Hemisphere. It is then within the scope of ACP. I have though some major issues with the current version of the paper that I recommend to be addressed before accepting the paper for publication.

a) This is a short paper and about 90\% of it is just a repetition of a MODEL analysis of a SSW (e.g. as in Tweedy et al., 2013). In this case the novelty is that it occurs in the SH but, as shown, the physics (dynamics) and chemistry is the same as in the NH. Apparently the new aspect is the more important role of O in the SH than in the NH

during SSWs (see the abstract). However, as discussed in text, it seems more related to the timing of the SSW along the winter rather than to the hemisphere itself. Then, I cannot see significant differences between the effects of SSWs in the NH and SH, a major motivation of the work. Therefore, the current version of the manuscript seems like an exercise and with very little NEW information.

As the manuscript is mainly based on model analyses, I would have expected (and I suggest to make it for the revised version) a more detailed analysis of the differences between the dynamical effects in both hemispheres. I think this would be interesting because the origin of GWs is substantially different in both hemispheres and this might have a different impact on the propagation and evolution of the SSWs effects. This would require model comparisons in both hemispheres.

b) The experimental "evidence" of SSW in mesospheric O3 in the SH from GOMOS is not very convincing. Fig. 6 shows that the O3 enhancement actually occurs 2 days BEFORE the response in the model, e.g. before the actual SSW. Also similar enhancements are seen at other days in GOMOS measurements which are not particularly correlated to SSWs. Personally, I am not convinced of such effect from Fig. 6.

This "experimental" evidence is relegated to just one page at the end of the paper. I would have written the manuscript the other way around. First show the evidence of the effect and then do the analysis. However the evidence is so weak that I doubt if it worth presenting it and then focus the manuscript only on model simulations.

In case the authors (or editor) decide to include the measurements, I would strongly recommend to include a point-by-point measurements/model comparison, by using model output at measurements geolocations. Not having WACCM data available at the time of writing should not be a reason for not performing an appropriate analysis. Such a comparison is fundamental for the paper, if including the measurements.

c) I convey with David Siskind's comment that the previous work by Cox et al. GRL,

2005, should be discussed and the model inter-comparison are highly recommended, particularly since the result on the hemispheric differences (SSW timing differences at the end) are based on WACCM model simulations.

Comments:

Abstract.

Lines 12-13. I would have written the sentence the other way around, e.g., model reproduces the measurements. Or change "demonstrate" for "show".

Lines 15-17. This sentence does not fully reflect the results. From this reading it seems that O/H plays a different role during SSWs in the Southern and Northern Hemispheres. However, as discussed in the manuscript and summarized in the Conclusions (lines 246-248), this seems more a question of the timing of the SSW along the winter rather than related to the hemisphere itself. I suggest to re-write the sentence. Also, state explicitly that it is a model result, i.e., not seen or being derived from the GOMOS measurements.

Lines 57-59. Note that the units of $c_1$ are different from those of $c_2$ and $c_3$. They are $cm^6\ molecules^{-2}\ s^{-1}$.

Lines 68-69. According to this statement I was then expecting to see the evidence (GOMOS measurements) first and its explanation with WACCM simulations later, not the other way around. I am not fully convinced of this "for the first time" $O_3$ enhancement during a SH SSW (see below).

Sec. 3.1

Related to previous comment, make clear from the beginning in this section and in the legend of the figures that the results shown in Figs. 1-4 are model simulations.

Line 177. Would then be better to use zonal mean of nighttime data only?

Lines 222-223. Sentence: "The largest enhancement is observed in the end of

[Figure]

September, when ..." But the largest peak in the measurements occurs 2 days BE-FORE the SW, on 23rd Sep !!

Line 229. About the sentence "This may also contribute to the lack of exact day-to-day correspondence with the WACCM-SD..." Correspondence with WACCM is very important since it is nudged so it should reflect very well the timing of the SSW. So this correspondence is crucial to attribute the causes to the effects and, in my opinion, the lack of such correspondence cast doubts on the evidence of the SSW effects on the mesospheric O3 in GOMOS. By the way, the drawing of the "Sep. 23" line in Fig. 6 is misleading since the SSW actually occurs on Sep 25 (see Fig. 5).

Lines 231-233. To make a direct comparison GOMOS/WACCM is essential to this study and to the confirmation of the O3 enhancement during this SSW. You should consider the temporal and geographical collocations of WACCM with GOMOS measurements in order to make the comparison meaningful and credible.

Lines 236-239. I do not agree with this sentence. In my opinion, in the best case, the detection of the O3 enhancement during the SSW by the presented GOMOS measurements is questionable.

Lines 246-248. Then, the attribution of the O3 changes in the studied SSWs in the SH and NH seems more related to the timing of the SSWs than to the hemisphere itself. This should be reflected in the abstract.

Figs. 1-4. Clarify that the plotted results are model calculations

Fig. 2. There is no "dash" line, is it the THICK GREY line?

Figs. 3 and 4. Typo, mesospause -> mesopause. Better say "around the mesopause"? (The mesopause is not defined by the 1e-3 hPa pressure level).

Fig. 5b. It might be more clear to show the effects of perturbing T, H or O on the O3 anomaly rather than O3 itself.

---

## Referee Comment (RC2) · Anonymous Referee #2 · 8 Dec 2016

First review of "Nighttime Mesospheric Ozone During the 2002 Southern Hemispheric Major Stratospheric Warming" by Smith-Johnsen for possible publication in ACP. The paper presents SSW effects on the secondary ozone maximum in the SH using Specified Dynamics version of WACCM and compares to GOMOS ozone measurements. These SH results are contrasted with NH SSW effects reported by Tweedy et al (2013). Increases in nighttime polar mesopause ozone are attributed to enhanced upwelling and cooling and "effects of atomic oxygen" (vs. hydrogen, as is the case in the NH). I have questions about the spatial regions chosen for the averages, concerns that the model output is not at the GOMOS profile locations, and left wondering how the results are different from and novel compared to Tweedy. The paper discussions are too brief,

and so the authors have the opportunity to expand on the significance of geographical distributions (if they are relevant this might be a possible novel result to emphasize?), and do a more thorough job of comparing temperature and trace gas distributions to other satellite observations (such as SABER as mentioned by Siskind). I think this paper has the potential for publication but not in its present form.

General Comments:

Need line numbers for the next review.

The intro is too general. Give brief motivation and literature review of SSW effects on the secondary ozone layer and why it is important to understand this coupling.

2. Methods, model and instrumentation – mention the accuracy of SD-WACCM mesospheric temperatures. I believe there are very large biases that depend on latitude and season. What are the biases in the Antarctic winter polar night jet above the nudging altitude? Cite others such as Garcia, Marsh, and other papers by Smith who go into more detail on model specifications.

Defend averaging 55-75 instead of 60-pole or 70-pole. Why an annulus instead of a polar cap area average?

Specific Comments:

Scherag-> Scherhag

This sentence doesn't make sense: "The main ozone source at this altitude is atomic oxygen, while the sinks are atomic hydrogen and atomic oxygen." Please check. Is atomic oxygen a net source or a net sink? What about molecular oxygen talked about in section 3.2?

Smith et al. (2014) -> should be 2015

Figure 1 – add figure panels a) – h) for reference. Figure 1 does not support this assertion, "After the onset, the PNJ recovery to eastward direction is only found above

[Figure]

10-1 hPa, while between this level and 101 hPa, westward winds persisted into the summer."

Figure 1 of w, T, O3: why average between 55-75 and not over the polar cap 75-pole? The anomalies the authors attribute to the 4 SSW pulses are not apparent. Averaging over the polar cap is more physically appropriate due to the meteorology (roughly vortex centered vertical motions, T, tracers) as well as providing more nighttime conditions. The current analysis is neither appropriate nor convincing.

Clarify "The zonal wind reversals mark strong ozone departures from the climatology in the mesopause region." – what does this mean? Change "mark" to "coincide with" ???

Figure 2: I wonder if averaging over the polar cap will result in more robust correlation between ozone and atomic oxygen? Give correlation coefficients between ozone and other variables. The statement "Atomic oxygen shows weak decreases, within the standard deviation of the climatology." Is not really supported due to the large fluctuations. What appears more clear is that overall ozone is elevated because overall (on average) hydrogen is depleted. In addition, the short term ozone enhancements during the SSW pulses appear to nicely coincide with episodic low hydrogen values. This seems consistent (not contrary) to the NH results of Tweedy.

Figures 3-4: The paragraph devoted to describing these figures is too brief. Can DIFFERENCES from 20 Sept to 25 Sept be shown to clarify where there are changes in the different fields? I think showing the "geography" of the ozone changes is a novel result but the discussion is lacking and the authors do not pursue this avenue. Figure 3 caption and text: specify how many days before the major SSW. But given the multiple warming pulses, isn't it really both before and after SSWs? Figures 3-4: consider combining these figures with F3 as the left column of 4 maps and F4 as the right column of 4 maps. Side-by-side would make the comparison easier. Why are these two particular days chosen? Are they representative? Increase ozone color bar range. Add

contours to all panels. Specify outer boundary (30S?). How well does the temperature map compare to SABER (or any other measurement of) temperature at these altitudes on these days? Can we trust daily geographic maps of SD-WACCM fields at these altitudes? If so, say so explicityly. If not, add a discussion including caveats.

Section 3.2 this is the first mention of molecular oxygen. Figure 5 – add figure panels a), b). In discussion of these results be explicit about comparisons to Tweedy. What is different and what is novel?

3.3 GOMOS satellite observations – move discussion of the instrument sampling and dataset gaps up to section 2.

Figure 6. I agree with reviewer 1 that this analysis should be SD-WACCM at the GOMOS profile locations (not spatial averages where GOMOS did not sample). This analysis needs to be re-done as it is critical to the paper. It should also be given more emphasis. Interpretation of results would be easier if the authors included GOMOS and SD-WACCM on the same plot. Is there agreement between the model and the observations? If so, what do both say about SSW impacts on the secondary ozone layer that are different from Tweedy? If not, why not and what does that mean in terms of the model capabilities.

---

## Author Comment (AC1) · 20 Jan 2017

Anonymous Referee #1 -

The manuscript addresses a topic of high scientific interest for knowing the state and variability of the mesosphere, in particular, the perturbations of the O3 secondary maximum distribution to a major SSW in the Southern Hemisphere. It is then within the scope of ACP. I have though some major issues with the current version of the paper that I recommend to be addressed before accepting the paper for publication.

General comments:

a) This is a short paper and about 90\% of it is just a repetition of a MODEL analysis

of a SSW (e.g. as in Tweedy et al., 2013). In this case the novelty is that it occurs in the SH but, as shown, the physics (dynamics) and chemistry is the same as in the NH. Apparently the new aspect is the more important role of O in the SH than in the NH during SSWs (see the abstract). However, as discussed in text, it seems more related to the timing of the SSW along the winter rather than to the hemisphere itself. Then, I cannot see significant differences between the effects of SSWs in the NH and SH, a major motivation of the work. Therefore, the current version of the manuscript seems like an exercise and with very little NEW information. As the manuscript is mainly based on model analyses, I would have expected (and I suggest to make it for the revised version) a more detailed analysis of the differences between the dynamical effects in both hemispheres. I think this would be interesting because the origin of GWs is substantially different in both hemispheres and this might have a different impact on the propagation and evolution of the SSWs effects. This would require model comparisons in both hemispheres.

AUTHOR REPLY: This is a good suggestion. We will do a more thorough comparison between this event and the Northern hemisphere mid-winter SSWs, and expand the discussion on interhemispheric and seasonal differences in the updated version of the paper.

b) The experimental "evidence" of SSW in mesospheric O3 in the SH from GOMOS is not very convincing. Fig. 6 shows that the O3 enhancement actually occurs 2 days BEFORE the response in the model, e.g. before the actual SSW. Also similar enhancements are seen at other days in GOMOS measurements which are not particularly correlated to SSWs. Personally, I am not convinced of such effect from Fig.6. This "experimental" evidence is relegated to just one page at the end of the paper. I would have written the manuscript the other way around. First show the evidence of the effect and then do the analysis. However the evidence is so weak that I doubt if it worth presenting it and then focus the manuscript only on model simulations. In case the authors (or editor) decide to include the measurements, I would strongly recommend

to include a point-by-point measurements/model comparison, by using model output at measurements geolocations. Not having WACCM data available at the time of writing should not be a reason for not performing an appropriate analysis. Such a comparison is fundamental for the paper, if including the measurements.

AUTHOR REPLY: We will move the GOMOS section to the beginning of the paper and use it as motivation for our study, rather than validation of model results, as you suggest. The GOMOS data does show higher values of ozone at about the same time and altitude as we see in WACCM, though the exact timing is not perfect. We have analysed our WACCM simulations with spatial (though not temporal) geolocation to GOMOS, but the results were not as clear due to the sparsity of GOMOS measurements, see Figure attached. The number of observations fluctuates around 10 per day in the week around the onset, with only 2 points on SEPT 23. The current model output is once a day, and we believe that doing a new model run with temporal geocolocation to GOMOS would not improve these collocation results much, since change in the GOMOS sampling in terms of local time means that there is a tidal component. We know that the tides in WACCM are unrealistically weak [Smith, 2012]. We did not plot GOMOS alongside WACCM since we know that there is a strong negative bias in WACCM. Smith, A. K., 2012. Global dynamics of the MLT. Surv. Geophys., DOI: 10.1007/s10712-012-9196-9

c) I convey with David Siskind's comment that the previous work by Cox et al. GRL, 2005, should be discussed and the model intercomparison are highly recommended, particularly since the result on the hemispheric differences (SSW timing differences at the end) are based on WACCM model simulations.

AUTHOR REPLY: The paper by Coy et al, 2005, will be referenced and discussed further, and a comparison will be done for the precursory mesospheric disturbances.

Abstract. Lines 12-13. I would have written the sentence the other way around, e.g., model reproduces the measurements. Or change "demonstrate" for "show".

AUTHOR REPLY: This sentence will be rewritten, and the paper restructured so GO-

[Figure]

MOS observations will be used as motivation and not as model validation.

Lines 15-17. This sentence does not fully reflect the results. From this reading it seems that O/H plays a different role during SSWs in the Southern and Northern Hemispheres. However, as discussed in the manuscript and summarized in the Conclusions (lines 246-248), this seems more a question of the timing of the SSW along the winter rather than related to the hemisphere itself. I suggest to re-write the sentence. Also, state explicitly that it is a model result, i.e., not seen or being derived from the GOMOS measurements.

AUTHOR REPLY: The conclusion will be rewritten, and a more thorough discussion of the interhemispheric differences will be included. We will also highlight better throughout the paper that this is a model study, and that all results are model results.

Lines 57-59. Note that the units of c1 are different from those of c2 and c3. They are cm6 molecules-2 s-1.

AUTHOR REPLY: Yes, you are of course correct. This will be corrected in updated version of draft.

Lines 68-69. According to this statement I was then expecting to see the evidence (GOMOS measurements) first and its explanation with WACCM simulations later, not the other way around. I am not fully convinced of this "for the first time" O3 enhancement during a SH SSW (see below).

AUTHOR REPLY: This sentence will be rewritten, and the paper restructured so GOMOS observations will be used as motivation and not as model validation.

Sec. 3.1. Related to previous comment, make clear from the beginning in this section and in the legend of the figures that the results shown in Figs. 1-4 are model simulations.

AUTHOR REPLY: We will highlight better throughout the paper that this is a model study, and that all results are model results.

Line 177. Would then be better to use zonal mean of nighttime data only?

AUTHOR REPLY: Yes, I agree. A zonal mean of only the nighttime data will be used, to avoid confusion.

Lines 222-223. Sentence: "The largest enhancement is observed in the end of September, when ..." But the largest peak in the measurements occurs 2 days BE-FORE the SW, on 23rd Sep !!

AUTHOR REPLY: The onset date used as a reference date corresponds to the wind reversal at 10 hPa, according to common practice in describing SSW; per se, it is not unanticipated that there could be precursory ozone fluctuations at 90 km. There is however disagreement between GOMOS observations and the model as mentioned above. The exact timing of the peaks in GOMOS will not be directly compared to the timing in WACCM, for the reason discussed above. This issue will be more thoroughly discussed.

Line 229. About the sentence "This may also contribute to the lack of exact day-to-day correspondence with the WACCM-SD..." Correspondence with WACCM is very important since it is nudged so it should reflect very well the timing of the SSW. So this correspondence is crucial to attribute the causes to the effects and, in my opinion, the lack of such correspondence cast doubts on the evidence of the SSW effects on the mesospheric O3 in GOMOS. By the way, the drawing of the "Sep. 23" line in Fig. 6 is misleading since the SSW actually occurs on Sep 25 (see Fig. 5).

AUTHOR REPLY: That sentence will be removed, and paper restructured so GOMOS observations will be used more as motivation than as model validation. The exact timing of the peaks in GOMOS will not be directly compared to the timing in WACCM. But the same reference dates will be drawn on all graphs, for readability.

Lines 231-233. To make a direct comparison GOMOS/WACCM is essential to this study and to the confirmation of the O3 enhancement during this SSW. You should
consider the temporal and geographical collocations of WACCM with GOMOS measurements in order to make the comparison meaningful and credible.

AUTHOR REPLY: We believe that doing a new model run with temporal geocolocation to GOMOS would not improve our results much, as the tidal component in WACCM is too weak, as discussed above.

Lines 236-239. I do not agree with this sentence. In my opinion, in the best case, the detection of the O3 enhancement during the SSW by the presented GOMOS measurements is questionable.

AUTHOR REPLY: Sentence will be removed, and the GOMOS data will be shown as vertical profiles, comparing 2002 to the two following years, to show that it is higher in 2002.

Lines 246-248. Then, the attribution of the O3 changes in the studied SSWs in the SH and NH seem

s more related to the timing of the SSWs than to the hemisphere itself. This should be reflected in the abstract.

AUTHOR REPLY: I agree, this can be rephrased. A more extensive comparison and discussion of hemispheric and seasonal differences will be included in updated paper.

Figs. 1-4. Clarify that the plotted results are model calculations

AUTHOR REPLY: We will highlight better throughout the paper that this is a model study, and that all results are model results.

Fig. 2. There is no "dash" line, is it the THICK GREY line?

AUTHOR REPLY: Thanks for highlighting, typing error will be fixed.

Figs. 3 and 4. Typo, mesospause -> mesopause. Better say "around the mesopause"? (The mesopause is not defined by the 1e-3 hPa pressure level).

AUTHOR REPLY: Thanks for highlighting, typing error will be fixed.

Fig. 5b. It might be more clear to show the effects of perturbing T, H or O on the O3 anomaly rather than O3 itself.

AUTHOR REPLY: We will explore this suggestion, and see how the results will look when using the anomaly rather than O3 itself.

Anonymous Referee #2 -

First review of "Nighttime Mesospheric Ozone During the 2002 Southern Hemispheric Major Stratospheric Warming" by Smith-Johnsen for possible publication in ACP. The paper presents SSW effects on the secondary ozone maximum in the SH using Specified Dynamics version of WACCM and compares to GOMOS ozone measurements. These SH results are contrasted with NH SSW effects reported by Tweedy et al (2013). Increases in nighttime polar mesopause ozone are attributed to enhanced upwelling and cooling and "effects of atomic oxygen" (vs. hydrogen, as is the case in the NH). I have questions about the spatial regions chosen for the averages, concerns that the model output is not at the GOMOS profile locations, and left wondering how the results are different from and novel compared to Tweedy. The paper discussions are too brief, and so the authors have the opportunity to expand on the significance of geographical distributions (if they are relevant this might be a possible novel result to emphasize?), and do a more thorough job of comparing temperature and trace gas distributions to other satellite observations (such as SABER as mentioned by Siskind). I think this paper has the potential for publication but not in its present form.

General Comments:

Need line numbers for the next review.

AUTHOR REPLY: I am sorry that you didn't get the version with line numbers.

The intro is too general. Give brief motivation and literature review of SSW effects on the secondary ozone layer and why it is important to understand this coupling.

AUTHOR REPLY: The introduction will be rewritten to include a better motivation, aim and scope of the paper.

2. Methods, model and instrumentation – mention the accuracy of SD-WACCM mesospheric temperatures. I believe there are very large biases that depend on latitude and season. What are the biases in the Antarctic winter polar night jet above the nudging altitude? Cite others such as Garcia, Marsh, and other papers by Smith who go into more detail on model specifications.

AUTHOR REPLY: The model section will be expanded to include more on model specifications and biases in this season and altitude region.

Defend averaging 55-75 instead of 60-pole or 70-pole. Why an annulus instead of a polar cap area average?

AUTHOR REPLY: We found that the highest ozone values were in this latitude band. But as you mention, a polar average would make more physical sense and make the comparison to [Tweedy, 2013] and [Smith, 2015] easier. We will check if a polar average highlights the results more clearly,

Specific Comments:

Scherag-> Scherhag

AUTHOR REPLY: Thanks for highlighting, typing error will be fixed.

This sentence doesn't make sense: "The main ozone source at this altitude is atomic oxygen, while the sinks are atomic hydrogen and atomic oxygen." Please check. Is atomic oxygen a net source or a net sink? What about molecular oxygen talked about in section 3.2?

AUTHOR REPLY: Atomic oxygen appears both in the source and the sink terms. Senteced will be rephrased for clarification.

Smith et al. (2014) -> should be 2015 Thanks for highlighting, typing error will be fixed.

Figure 1 – add figure panels a) – h) for reference. Figure 1 does not support this assertion, "After the onset, the PNJ recovery to eastward direction is only found above $10^{-1}$ hPa, while between this level and $10^{1}$ hPa, westward winds persisted into the summer."

AUTHOR REPLY: a) – h) panels will be added. Senteced will be rephrased for clarification.

Figure 1 of w, T, O3: why average between 55-75 and not over the polar cap 75-pole? The anomalies the authors attribute to the 4 SSW pulses are not apparent. Averaging over the polar cap is more physically appropriate due to the meteorology (roughly vortex centered vertical motions, T, tracers) as well as providing more nighttime conditions. The current analysis is neither appropriate nor convincing.

AUTHOR REPLY: As discussed above.

Clarify "The zonal wind reversals mark strong ozone departures from the climatology in the mesopause region." – what does this mean? Change "mark" to "coincide with" ???

AUTHOR REPLY: ЗńCoincide withЗż will be a more correct phrasing, yes. The sentence will be rephrased.

Figure 2: I wonder if averaging over the polar cap will result in more robust correlation between ozone and atomic oxygen? Give correlation coefficients between ozone and other variables. The statement "Atomic oxygen shows weak decreases, within the standard deviation of the climatology." Is not really supported due to the large fluctuations. What appears more clear is that overall ozone is elevated because overall (on average) hydrogen is depleted. In addition, the short term ozone enhancements during the SSW pulses appear to nicely coincide with episodic low hydrogen values. This seems consistent (not contrary) to the NH results of Tweedy.

AUTHOR REPLY: All figures will be changed to polar cap average. Including correlation

coefficients is a good idea, this will be done. We will do a more thorough comparison to the Northern hemisphere, and expand the discussion on interhemispheric and seasonal differences in the updated version of the paper.

Figures 3-4: The paragraph devoted to describing these figures is too brief. Can DIF-FERENCES from 20 Sept to 25 Sept be shown to clarify where there are changes in the different fields? I think showing the "geography" of the ozone changes is a novel result but the discussion is lacking and the authors do not pursue this avenue. Figure 3 caption and text: specify how many days before the major SSW. But given the multiple warming pulses, isn't it really both before and after SSWs? Figures 3-4: consider combining these figures with F3 as the left column of 4 maps and F4 as the right column of 4 maps. Side-by-side would make the comparison easier. Why are these two particular days chosen? Are they representative? Increase ozone color bar range. Add contours to all panels. Specify outer boundary (30S?). How well does the temperature map compare to SABER (or any other measurement of) temperature at these altitudes on these days? Can we trust daily geographic maps of SD-WACCM fields at these altitudes? If so, say so explicityly. If not, add a discussion including caveats.

AUTHOR REPLY: Yes, the difference from 20 to 25 will be a better and clearer way to show the change, this will be including in the updated draft. Looking further at the geographical distribution is a great idea, this will be further investigated and the section will be expanded.

Section 3.2 this is the first mention of molecular oxygen. Figure 5 – add figure panels a), b). In discussion of these results be explicit about comparisons to Tweedy. What is different and what is novel?

AUTHOR REPLY: Figure panels will be added, thanks. The comparison with NH and the Tweedy study will be expanded in the discussion section, and the differences and similarities will be highlighted in a clearer way.

3.3 GOMOS satellite observations – move discussion of the instrument sampling and
dataset gaps up to section 2.

AUTHOR REPLY: The GOMOS section will be moved from the discussion/validation section, to the introduction, and used more as a motivation. A discussion of the sampling will be included.

Figure 6. I agree with reviewer 1 that this analysis should be SD-WACCM at the GO-MOS profile locations (not spatial averages where GOMOS did not sample). This analysis needs to be re-done as it is critical to the paper. It should also be given more emphasis. Interpretation of results would be easier if the authors included GOMOS and SD-WACCM on the same plot. Is there agreement between the model and the observations? If so, what do both say about SSW impacts on the secondary ozone layer that are different from Tweedy? If not, why not and what does that mean in terms of the model capabilities.

AUTHOR REPLY: We will move the GOMOS section to the beginning of the paper and use it as motivation for our study, rather than validation of model results, as you suggest. The GOMOS data does show higher values of ozone at about the same time and altitude as we see in WACCM, though the exact timing is not perfect. We have analysed our WACCM simulations with spatial (though not temporal) geolocation to GOMOS, but the results were not as clear due to the sparsity of GOMOS measurements, see Figure attached. The number of observations fluctuates around 10 per day in the week around the onset, with only 2 points on SEPT 23. The current model output is once a day, and we believe that doing a new model run with temporal geocolocation to GOMOS would not improve these collocation results much, since change in the GOMOS sampling in terms of local time means that there is a tidal component. We know that the tides in WACCM are unrealistically weak [Smith, 2012]. We did not plot GOMOS alongside WACCM since we know that there is a strong negative bias in WACCM.

D. Siskind -

This is an interesting paper and it is good to revisit this particular event which was,

and still is, quite unusual. My main comment is that I think they could and should do a bit more to validate their model. One paper that is probably of relevance is that of Coy et al., (2005) "Modeling the 2002 minor warming event", [GRL, L 07808, doi: 10.1029/2005GL022400]. That presented what, to my knowledge, remains the only simulation of the mesospheric response to the SH 2002 warming. I think this should be cited. Admittedly, they covered the minor warming precursor in August, not the major event in September. This is because SABER yawed away from the SH and did not observe the major event. Nonetheless, the August period is covered by the present authors' simulations and figures. For reference, I have attached Figure 3 from the Coy paper. Note that Coy et al show the vertical structure of the event up to the upper mesosphere. I suggest it would be useful for a similar analysis to be done here to help validate their modeled mesospheric response.

AUTHOR REPLY: We are grateful for pointing to us the Coy paper; it will be referenced and discussed. We will refer in details to the papers by Coy (2005) and also Smith et al. (2015), who have already looked at SABER data for the same period and with some focus on the southern high latitudes. In particular, Smith et al. examines SABER O3, O, T, H for a series of years, including 2012, and makes a comparison with SD-WACCM.

My other question concerns the hydrogen. They refer to a climatology in their figure. Is this climatology from WACCM itself? Because SABER did measure hydrogen during the August event. A paper by Mlynczak et al, JGR, 2014 documents the SABER H product. I believe this is an underutilized dataset and the present paper offers an ideal opportunity to compare their model with data.

AUTHOR REPLY: All figures showed are model data, except for the GOMOS-related Figure 6. This can be highlighted better in the updated version of the paper. Since Smith et al.(2015) already examined SABER H for a series of years, including 2002, with a focus on the high latitudes, and made a comparison with SD-WACCM, we will refer to the figures in that paper in so far as possible.

Finally, and I'm sure they've noticed already, but somebody misspelled stratospheric in their title. (start) It seems correct on the PDF, but incorrect on the web site

AUTHOR REPLY: Thanks for highlighting, this has been corrected.

[Figure]

**GOMOS O3 - 2002 - [55;75]deg South**

572 measurements over 31 operating days

between 01-Aug-2002 and 30-Sep-2002

Number of Nighttime Measurements

**Fig. 1.**

[Figure]

**Fig. 2.**